# Modified Fluoroquinolones as Antimicrobial Compounds Targeting *Chlamydia trachomatis*

**DOI:** 10.3390/ijms23126741

**Published:** 2022-06-16

**Authors:** Thi Huyen Vu, Erika Adhel, Katarina Vielfort, Ngûyet-Thanh Ha Duong, Guillaume Anquetin, Katy Jeannot, Philippe Verbeke, Sofia Hjalmar, Åsa Gylfe, Nawal Serradji

**Affiliations:** 1University of Engineering and Technology, Vietnam National University, Hanoi (VNUH), Vietnam; huyenbionano@gmail.com; 2Université Paris Cité, CNRS, ITODYS, F-75013 Paris, France; erikaadhel@gmail.com (E.A.); thanh.haduong@u-paris.fr (N.-T.H.D.); guillaume.anquetin@u-paris.fr (G.A.); 3Department of Molecular Biology, Umeå University, SE-901 87 Umeå, Sweden; katarina.vielfort@umu.se; 4Centre National de Référence de la Résistance aux Antibiotiques, Centre Hospitalier Universitaire de Besançon, F-25030 Besançon, France; katy.jeannot@univ-fcomte.fr; 5Chrono-Environnement, UMR 6249, CNRS Faculté de Médecine-Pharmacie, Université Bourgogne-Franche Comté, F-25000 Besançon, France; 6Faculté de Médecine Xavier Bichat, Université Paris Cité, INSERM U1149, F-75018 Paris, France; philippe.verbeke@inserm.fr; 7Department of Clinical Microbiology, Umeå University, SE-901 85 Umeå, Sweden; sofia.hjalmar@hotmail.com

**Keywords:** antibacterial, bactericidal, inhibitors, 8-hydroxyquinoline, iron

## Abstract

*Chlamydia trachomatis* causes the most common sexually transmitted bacterial infection and trachoma, an eye infection. Untreated infections can lead to sequelae, such as infertility and ectopic pregnancy in women and blindness. We previously enhanced the antichlamydial activity of the fluoroquinolone ciprofloxacin by grafting a metal chelating moiety onto it. In the present study, we pursued this pharmacomodulation and obtained nanomolar active molecules (EC_50_) against this pathogen. This gain in activity prompted us to evaluate the antibacterial activity of this family of molecules against other pathogenic bacteria, such as *Neisseria gonorrhoeae* and bacteria from the ESKAPE group. The results show that the novel molecules have selectively improved activity against *C. trachomatis* and demonstrate how the antichlamydial effect of fluoroquinolones can be enhanced.

## 1. Introduction

*Chlamydia trachomatis* (*C. trachomatis*), a Gram-negative bacterium, is the pathogen responsible for a sexually transmitted bacterial infection (STI) affecting more than 200 million people per year [1] and the eye infection, trachoma [2]. Unless treated promptly, sexually transmitted *chlamydia* can lead to infertility and ectopic pregnancy in women and ocular and respiratory infections in newborns. Trachoma is the cause of blindness in 1.9 million people [2] and over 30 million people were treated with antibiotics in 2020 to control the infection [3].

*C. trachomatis* is a strict intracellular bacterium. Therefore, it can only multiply in a parasitophorous inclusion located in the cytosol of a eukaryotic cell. Intracellular *C. trachomatis* is protected from antibiotics that fail to reach sufficiently high intracellular concentrations, leaving few treatment options available [4].

Iron acquisition is a key step during *C. trachomatis* development in the host cell [5,6]. Indeed, the growth of this pathogen is iron-dependent [6,7,8]. In vitro and in vivo studies have shown that iron chelators of the acylsalicylidene hydrazone type have an effect against *Chlamydia*, with a minimum bactericidal concentration (MBC) of 25–50 μM [9]. This inhibition is removed when iron (iron sulfate or chloride, holotransferrin and hololactoferrin) is added to the medium. Moreover, this inhibition appears to be specific to iron, as the introduction of other metal cations (Ca^2+^, Mn^2+^, Mg^2+^ and Zn^2+^) does not affect the growth of *C. trachomatis*.

We were interested in the relationship between the bacterial iron requirements and the possibility of inhibiting the growth of this bacterium [7,8,10]. In this context, we first synthesized compounds with a central 4-amino-3-isoxazolidinone nucleus, since D-cycloserine, a 3-isoxazolidonone derivative, is active against *C. trachomatis*, but with an unknown mechanism of action. In order to reach compounds more potent than D-cycloserine, we grafted onto it known iron-chelating moieties, such as catechol or 8-hydroxyquinoline groups, to deplete iron from the bacterial environment and successfully obtained active compounds [11]. Furthermore, with the aim of improving the moderate antichlamydial activity of ciprofloxacin, a potent, broad-spectrum commercially available antibiotic, we prepared a ciprofloxacin conjugate **1** using 8-hydroxyquinoline to try to obtain compounds with dual activity: iron chelation and antibiotic efficiency (Figure 1) [12]. The conjugate **1** was found to be more active than ciprofloxacin [13]. These results have opened up the possibility of preparing new derivatives of ciprofloxacin, even more active than the parent antibiotic, which would complete the therapeutic arsenal available to fight this bacterium.

In line with our previous results, we modified compound **1**‘s structure in order to improve its antichlamydial activity. For this, we considered modifying the lipophilicity of the compounds, their global flexibility or their ability to complex iron. Indeed, we: (i) lengthened the carbon chain via its hydroxyl function; (ii) replaced the amide moiety, known to confer structural rigidity and resistance to hydrolysis with a more flexible methylene group; (iii) modified the position of the antibiotic on the 8-hydroxyquinoline heterocycle to affect its iron-chelating potency; (iv) assessed other fluoroquinolones never tested against this pathogen for their putative antichlamydial activity.

Moreover, with STIs, chlamydial and gonococcal infections often coexist. *Neisseria gonorrhoeae* (gonococci) is a bacterial pathogen responsible for gonorrhoea, the second most common bacterial STI after *C. trachomatis*, which is also responsible for infertility and blinding eye infections [1]. *N. gonorrhoeae*, previously readily curable with many classes of antibiotics, including ciprofloxacin [14], is now considered as a multidrug-resistant organism due to its decreasing susceptibility to last-resort treatments and increasing difficulty in infection treatment [15]. The gain in bactericidal activity observed against *C. trachomatis* with compound **1**, compared to that of the parent antibiotic, ciprofloxacin, encouraged us to look at the antigonococcal activity of this family of compounds.

Finally, the broad-spectrum antibiotic activity of fluoroquinolones led us to evaluate the ability of these compounds to inhibit bacteria from the ESKAPE group, pathogenic bacteria frequently involved in healthcare-associated infections that are difficult to treat (*E. faecium*, *S. aureus*, *K. pneumoniae*, *A. baumannii*, *P. aeruginosa and Enterobacter species*).

## 2. Results and Discussion

### 2.1. Organic Synthesis

To increase the antibacterial activity of the previously published compound **1**, we synthesized several derivatives following different approaches. First, we alkylated the hydroxyl function on the quinoline heterocycle in order to increase lipophilicity (Figure 1). Then, we prepared an isomer of compound **1**, modifying the position of the covalent bond between the antibiotic and the quinoline moiety, from position C-2 to C-7 (Figure 2). We also replaced the latter using indole, another nitrogen heterocycle (Figure 2). Finally, we synthesized two fluoroquinolones already published but with unknown antichlamydial activity, 8-methoxyciprofloxacin [16] and desmethylofloxacin (Figure 3) [17].

Several ciprofloxacin derivatives (R = Me, Pr, Bu, Bn) were prepared, as presented in Figure 1. This strategy successively includes an alkylation of the 2-methylquinolin-8-ol, an oxidation of the methyl group to the corresponding carboxylic acid and the coupling of the latter with ciprofloxacin.

We adapted the procedures described by Qu et al. [18] or Terazzi et al. [19] to obtain compounds **2b–d**. For this, we introduced potassium iodide during the reaction of 2-methylquinolin-8-ol with the appropriate halides, using a base (K_2_CO_3_) in *N*,*N*-dimethylformamide (DMF) or acetone to afford the alkyl derivatives in **2b–d** good yields (89–99%).

Oxidation using selenium dioxide in 1,4-dioxane [18,19] afforded compounds **3b–d**. A first attempt was made to use potassium permanganate to oxidize aldehydes **3b,c** following the work of Qu et al. [18] without success. The alternative use of potassium hydrogen persulfate (Oxone^®^) [20] provided acids **4b** and **4c** with yields of 52% and 63%, respectively. The yield was increased to 91% when a mixture of sulfamic acid (H_2_NSO_3_H) and sodium chlorite (NaClO_2_) was used to obtain **4d**.

Peptide coupling of the commercially available 8-methoxyquinoline-2-carboxylic acid or compounds **4b–d** to ciprofloxacin led to compounds **5a–d** with good yields (71–92%). This reaction was carried out using *O*-(benzotriazol-1-yl)-*N*,*N*,*N′*,*N*′-tetramethyluronium tetrafluoroborate (TBTU) and *N*,*N*-diisopropylethylamine (DIEA).

According to the work published in 2020 where we demonstrated the important link between the carbonyl group position of the 8-hydroxyquinoline moiety and its iron-chelating properties [21], we synthesized compound **6** (Figure 2) using ciprofloxacin and 8-hydroxyquinoline-7-carboxylic acid with TBTU and DIEA at 80 °C.

The introduction of an indole instead of the 8-hydroxyquinoline moiety was also evaluated. Compounds are presented in Figure 2; these were obtained from the corresponding carboxylic acid and ciprofloxacin, as previously described for compound **1**, except for **10**. To get this compound, indole was first acylated with oxalyl chloride before undergoing a reaction with ciprofloxacin to provide the final compound [22].

The purpose of the synthesis of compound **14** was to study the influence of the modification of the coupling position of 8-hydroxyquinoline with ciprofloxacin. The synthesis of **14** is outlined in Figure 3. Following the protocol described by Yang et al. [23], commercially available 8-hydroxyquinoline was first chloromethylated to give 5-chloromethyl-8-hydroxyquinoline **13**. Then, **13** was reacted with ciprofloxacin to give compound **14**. In water, the reaction gave a yield of 37% and reached 84% when DMF was used instead.

Iron-chelating and antibacterial properties of the compounds synthesized were then investigated.

### 2.2. Iron-Chelating Properties

To study the putative link between the iron-chelating properties of this family of compounds and their enhanced antibacterial activity, we used absorption spectrophotometry. As compounds **5a–d** present an alkylated hydroxyl function in the quinoline moiety, they are not able to complex iron through this entity. We confirmed in SI that **5a**, as a model, only presents the previously described fluoroquinolone-chelating properties (i.e., through ciprofloxacin carboxylate and keto groups) [13].

Compounds **1** and **6** were compared according to their ability to complex iron (III). Complexation experiments were performed in a H_2_O/DMSO (1:1; *v*/*v*) mixture.

In theory, three potential sites for metal complexation can be identified in compound **6**: (i) between the carboxylate and the keto groups on the ciprofloxacin moiety (fluoroquinolone); (ii) between the hydroxyl and the carbonyl groups; or (iii) the nitrogen and hydroxyl functions on the 8-hydroxyquinoline part.

At pH 7, during iron exchange experiments between the Fe-nitrilotriacetic complex (Fe-NTA) and compound **6**, two isosbestic points at 256 and 342 nm and two ligand-to-metal charge-transfer (LMCT) bands appear at 435 nm and 570 nm (Figure 4). According to the plots of absorbance at these wavelengths, a single iron complex with a stoichiometry of one metal for one ligand is formed. In contrast to what was observed for compound **1**, two LMCT bands exist when iron is present and these bands are blue-shifted when the pH value increases, as already reported for hydroxyquinoline derivatives [13].

In order to compare the chelation mode of compounds **1** and **6**, the corresponding LMCT bands, observed at pH 7, are listed in Table 1 below.

In general, a similar mode of chelation most often results in analogous spectral modifications, such as the appearance of similar or even identical LMCT bands. We previously compared compound **1**’s LMCT bands with that of ciprofloxacin to conclude that compound **1** probably complexes the metal through its fluoroquinolone entity rather than through its 8-hydroxyquinoline group [13]. The spectral behavior of compound **6** is different, as two LMCT bands are observed at this pH. Moreover, the corresponding wavelengths are close to those observed when iron is added to a solution of 8-hydroxyquinoline (**HQ**, Table 1). Thus, compound **6** appears to chelate iron through its hydroxyquinoline part rather than through its fluoroquinolone moiety.

In an attempt to quantify the results, the absorption spectra (Figure 3) were analyzed by SPECFIT, which gives exchange constants for FeNTA and the Fe^3+^-**1** or Fe^3+^-**6** complexes (log K_11_). The calculated values of 4.23 ± 0.09 and 5.61 ± 0.15 obtained for compound **1** and **6**, respectively, suggest a higher affinity of compound **6** for the metal.

### 2.3. Antibacterial Activity

Compound **1**’s derivatives were investigated for antichlamydial activity in HeLa cells infected with *C. trachomatis* serovar L2 454/Bu. We first determined the MIC of all compounds and, due to technical limitations, MIC was defined as at least 95% growth inhibition.

Several of the novel analogs were more potent than ciprofloxacin, which was used as reference compound (Table 2).

The results highlight the following elements: the alkylation of the 8-hydroxyquinoline is favorable to the activity (**5a**, **5b**, **5d**), unlike replacing 8-hydroxyquinoline with an indole motif (compounds **7–12**).

In parallel, we evaluated the ability of two fluoroquinolones, **15**, 8-methoxyciprofloxacin [24] and **16**, desmethylofloxacin, with the previously unknown antichlamydial activity (Figure 4).

**15** was more efficient (MIC = 0.125 µM) in inhibiting chlamydial growth than ciprofloxacin (MIC = 0.5 µM). We hypothesize that the introduction of the 8-hydroxyquinoline motif on **15** may result in even more potent compounds.

In contrast, the weak activity of **16** suggests the importance of the methyl group found in ofloxacin, a fluoroquinolone with potent antichlamydial activity (Figure 4) [25]. Others have shown that newer fluoroquinolones have further improved activity against *C. trachomatis*, since sitafloxacin and tosufloxacin are more potent than ciprofloxacin and levofloxacin (the optical S-(-) isomer of ofloxacin) [26] and moxifloxacin is more potent than ofloxacin [27]. Our data further validated the idea that pre-existing fluoroquinolones can still be modified to improve their activity against *C. trachomatis*.

The most potent compounds (**1**, **5a**, **5b**, **5d**, **6**, **14** and **15**) were selected for further study against *C. trachomatis*. We measured their effects on progeny reinfection and obtained their MBC and 50% effective concentrations (Table 3) using *Chlamydia trachomatis* serovar L2 454/Bu, expressing the fluorescent red protein mCherry and detected without immunostaining [28]. The MBC value was similar to the MIC value for each compound (Table 1), with, at the most, one dilution step difference, confirming the bactericidal mode of action of these analogs. The EC_50_ values were in the nanomolar range and distinguished the activities of the different compounds more in detail (Table 3). The most potent compound was **6**, which had an EC_50_ of 13 nM.

In drug design, compound lipophilicity is a key parameter during early preclinical development [29]. As *C. trachomatis* is an intracellular bacterium, efficient antimicrobials must be able to cross several hydrophobic biological membranes (host cell, bacterial inclusion and bacterial wall membranes). The important ability of macrolides, such as azithromycin, to concentrate within host cells is probably related to their lipophilic behavior and their usefulness against *C. trachomatis* infection. Lipophilicity is often expressed as a partition coefficient (log P), which expresses how much a molecule dissolves in two non-miscible compartments: an organic/hydrophobic compartment and water. In our study, to establish the possible link between this property and the antibacterial activity of the molecules synthesized, cLog P was calculated with ChemDraw Ultra 13.0.0.3015 software.

Surprisingly, for the alkylated molecules, **5a**, **5b** and **5d**, no positive correlation is observed between their antibacterial activities and their calculated lipophilicities (Table 3), suggesting that other parameters are implicated in their good antibacterial activities.

We then tested antimicrobial compounds both in the presence (200 µM) and absence of iron (III) citrate to see if excess iron reduced the effect of the compounds against chlamydial infection. Indeed, several compounds, through their 8-hydroxyquinoline moiety, exhibit cation-chelating properties. Compounds belonging to the indole series **7**–**12** were not tested as they were not believed to complex iron through their indole frameworks. Results presented in Table 4 indicate that there is a difference only for **6** in the inhibition of *C. trachomatis* infection depending on the addition of iron. It is interesting to observe that **6** is the most potent compound in the absence of an excess of iron and the least potent in the presence of an excess of this metal. We demonstrated that a carbonyl group positioned on carbon 7 of the 8-hydroxyquinoline moiety leads to high iron-chelating properties [21]. **6** is expected to be a more potent iron chelator than **1**, its C2-isomer, at pH 7.4, and it is, therefore, more susceptible to modulations in the iron concentration in its environment. This experimental observation suggests that either the mechanism of action of this molecule is dual, with both iron deprivation of the cell and fluoroquinolone activity, or the **6**/iron(III) complex has a reduced capacity to reach its target and exert its inhibitory action.

A visualization of the immunostained *Chlamydia* inclusions (red immunostaining) treated with **6** at 0.125 µM is shown in Figure 5. The difference between the two images demonstrates that excess iron (III) citrate reduces the effect of **6**, as illustrated by the large number of visible *Chlamydia* inclusions.

To estimate the cell toxicity of the most potent compounds against HeLa cells, a neutral red uptake assay was performed. The results are shown in Table 5 as cell viability in percent relative to the DMSO-treated control. Table 5 indicates that all compounds except **12** show low toxicity towards HeLa cells at 10 µM.

The new generation fluoroquinolone, sitafloxacin, has improved its activity against *N. gonorrhoeae* compared to ciprofloxacin, and has also overcome ciprofloxacin resistance in this bacterium [30]. We were interested in whether the novel antimicrobial compounds in this study also had improved activity against *N. gonorrhoeae* and evaluated their potency against different strains. The WHO reference strains, WHO P (susceptible to ciprofloxacin) and WHO G (low-level resistance to ciprofloxacin), were used. MIC and MBC values are presented in Table 2 and show that ciprofloxacin remains the most effective compound against this pathogen. We can note that compound **14** exhibits significant activity and that it is the only one to have a methylene-type linker between the hydroxyquinoline entity and the fluoroquinolone moiety. *N. gonorrhoeae* is dependent on iron for its growth and we added 200 µM iron (III) citrate to see if it reduced the effect of 8-hydroxyquinoline-containing compounds on this bacterium. In contrast to *C. trachomatis*, excess iron (III) did not change the MIC of **6** (0.5 µM). Thus, the **6**-iron (III) complex probably accesses the molecular target in *N. gonorrhoeae*, as well as in **6** alone. *N. gonorrhoeae* has high-affinity systems for iron acquisition, for example, by the uptake of hematin [31] that is included in the culture medium. The fastidious broth used for *N. gonorrhoeae* culture also contained 0.13 µM iron (II) sulfate and the iron deprivation effect of **6** may be overridden by effective iron acquisition by *N. gonorrhoeae*. Compounds **1**, **5a** and **14** had lower MICs than **6** and were also not affected by excess iron (III) citrate.

The ciprofloxacin-resistant strain WHO-G was cross-resistant to all the compounds tested, indicating that the fluoroquinolone mode of action was retained for all and that these modifications did not overcome the resistance mechanisms.

Regarding bacteria from the ESKAPE group, experiments were first carried out on different bacterial species frequently encountered in human infections. As shown in Table 6 and Table 7, some compounds have interesting micromolar activities against *Staphylococcus aureus* (**1**, **5a**, **6**) and *E. coli* (**1**, **5a**). However, as for *N. gonorrhoeae*, the ciprofloxacin-resistant *Staphylococcus aureus ATCC700699* was cross-resistant to these new molecules. These results show that, although it is essential to identify new compounds active against the bacteria responsible for nosocomial infections, obtaining such compounds by modifying current antibiotic classes remains a challenge.

## 3. Materials and Methods

### 3.1. Organic Synthesis

#### 3.1.1. Materials and Methods

All commercially available reagents were used without further purification. To control the progress of a reaction, thin-layer chromatography was performed on plastic TLC sheets of silica gel 60 F254 (layer thickness 0.2 mm) from Merck. The structures of all compounds were verified by IR, ^1^H and ^13^C NMR spectra and HRMS. A Kofler Hot Bench (Wagner Munz™) with a wide temperature range (+50 °C to + 260 °C) was used for melting point determination. IR spectra were recorded on a PerkinElmer Spectrum 100 FT-IR spectrometer and NMR spectra were recorded in an appropriate deuterated solvent on a Bruker AC 400 spectrometer at 400.15 MHz for ^1^H and 100.62 MHz for ^13^C. The chemical shifts, in ppm, are referenced to the residual solvent signal. Coupling constants (*J*) are given in Hertz (Hz), chemical shifts in ppm and peak multiplicities are designated as usual. High-resolution mass spectra (HRMS) were recorded on a Waters spectrometer using electrospray ionization-TOF (ESI-TOF; Waters, Guyancourt, France) at the Centre de recherche de Gif, Équipe de Spectrométrie de Masse, Institut de Chimie des Substances Naturelles (Gif-sur-Yvette, France).

#### 3.1.2. Experimental Procedures

The experimental synthesis protocols are available in the Appendix A section of this manuscript.

### 3.2. Metal Chelation

#### 3.2.1. Stock Solutions

Solutions of compound **1**, or its derivatives, at 10^−2^ M in DMSO were prepared first. A 1 × 10^−4^ M solution of **1** (or its derivatives) in H_2_O/DMSO mixture (1:1, *v*/*v*) in HCl at pH 2 or in HEPES buffer (50 mM HEPES, 150 mM KCl) at pH 7 was used to perform the complexation experiments. FeCl_3_ and FeNTA solutions were prepared in acidic media (pH 2) or as previously described in [32], respectively.

#### 3.2.2. Spectrophotometric Measurements

Affinity constants [33] were determined by means of the SPECFIT32 Global Analysis program from measurements performed on a Cary 4000 spectrophotometer at 25.0 ± 0.5 °C.

### 3.3. Antibacterial Activity

#### 3.3.1. *C. trachomatis*

Cell culture and Chlamydia propagation

HeLa 229 cells (CCL-2.1; ATCC) were cultured at 37 °C (5% CO_2_) in a RPMI-1640 medium (HyClone) with 25 mM HEPES and 2 mM L-glutamine, supplemented with 10% fetal bovine serum (FBS) (Sigma-Aldrich, St Louis, MI, USA). *C. trachomatis* serovar L2 454/Bu (VR902B; ATCC) or transformed *C. trachomatis* serovar L2 454/Bu (VR902B; ATCC) expressing the fluorescent red protein mCherry (CT L2 mCherry) [28] were cultured in HeLa cells. *C. trachomatis* elementary bodies were purified, as described by Caldwell et al. [34], and stored at −80 °C in SPG (10 mM sodium phosphate, 5 mM L-glutamic acid and 0.25 M sucrose).

To determine the minimum inhibitory concentrations (MIC) of the compounds, 15,000 HeLa cells were seeded in flat-bottomed 96-well plates (Falcon) and infected 24 h later with *C. trachomatis* L2 454/Bu in HBSS (Hank’s Buffered Saline Solution) at a multiplicity of infection (MOI) of 0.25. After one hour of incubation at 37 °C (5% CO_2_), HBSS was replaced by a cell culture medium containing compounds in two-fold dilution and 0.5% DMSO. Compounds were tested in triplicate. The cells were fixed with methanol after 18 h and immunostained, as previously described, with a polyclonal rabbit anti-EB antibody [35], Alexa fluor 647-conjugated donkey anti-rabbit IgG (Jackson ImmunoResearch) and 1 µM 4′,6-diamidino-2-phenylindole (DAPI). *C. trachomatis* inclusion-forming units (IFU) were quantified with a Cytation 5 automated microscope (Agilent, Santa Clara, CA, USA) using a 4 X objective, ex 377 nm, em 447 nm for DAPI and ex 628 nm and em 685 nm for Alexa fluor 647. All visual fields were also manually inspected to ensure valid enumerations and intact cell layers. MIC was defined as the concentration where the number of IFUs was reduced by 95% or more compared to the DMSO control. The results were confirmed in at least two experiments run on different days. Selected compounds were also tested with 200 µM Fe(III) citrate added to the cell culture medium. For determination of minimum bactericidal concentration (MBC) and bactericidal EC_50_, 22,000 HeLa cells were infected with mCherry expressing *C. trachomatis* at a MOI of 0.5, and compounds were tested in triplicate wells with a final concentration of 0.2% DMSO [36]. *Chlamydia* progenies were harvested 46 h post-infection through osmotic host cell lysis by the addition of cold Milli-Q water; to get a final concentration of 1X SPG, 4X SPG was then added. The harvested bacteria were diluted in HBSS in 10-fold serial dilutions, added to fresh HeLa cells and incubated at 37 °C (5% CO_2_). One hour post-infection, HBSS was replaced with cell culture medium. At 44 h post-infection, Hoechst 33342 (Thermo Scientific, Rockford, IL, USA), 0.1 µg per well, was added 20 min prior to fixation with 4% formaldehyde (Sigma-Aldrich) for 15 min. The 96-well plates were analyzed by Arrayscan automated microscopy (ArrayScan VTI HCS, Thermo Scientific). Images were acquired in 10 visual fields with the 10× objective (Hoechst-ex386/23 nm, mCherry: ex549/15 nm) and the built-in software was used to count IFUs. Small, underdeveloped inclusions and red fluorescent artefacts were removed with a size cut-off. The latter was determined based on *Chlamydia* inclusions in DMSO control wells. Manual image inspection made it possible to dismiss any field that had lost cells during the washing process. Relative infection was expressed as the percentage of IFUs compared to DMSO-treated controls. All experiments were performed three times (three technical replicates in each experiment). MBC was defined as the compound concentration that reduced the infection to <0.1% relative to the DMSO-treated control infection. EC_50_ was calculated using GraphPad Prism 5. Values from three individual experiments were normalized by setting DMSO-treated infections to 100% for each experiment and calculating the relative infection for each compound concentration (Appendix A). Transformation of the concentrations to the log scale provided a non-linear regression. Curve fit was then performed using log(inhibitor) vs. normalized response−variable slope.

Cell viability assay

HeLa 229 cells were seeded, 10,000 cells per well, in 96-well plates (Falcon) in RPMI, supplemented as above with 5% heat-inactivated FBS. After 24 h, the cell culture medium was replaced with 100 µL new cell culture medium containing test compounds with 0.4% DMSO. The cells were then incubated with test compounds in triplicate at 37 °C with 5% CO_2_ for 21 h before the cell culture medium was removed. Neutral red staining was performed essentially as previously described [37]. Neutral red (0.33%) in 200 µL pre-warmed and sterile-filtered neutral red medium was added for 3 h at 37 °C with 5% CO_2_ to stain the lysosomes of viable cells. The cells were washed with 200 µL DPBS, and 100 µL neutral red de-stain solution (51% deionized water, 48% ethanol (100%) and 1% glacial acetic acid) was added. The 96-well plates were covered in metal foil and placed on a shaker for 20–30 min; absorption was measured at 540 nm. The cell viability was expressed in percent relative to the DMSO control.

#### 3.3.2. *N. gonorrhoeae*

*N. gonorrhoeae*, WHO-P and WHO-G [38], were subcultured on GC agar plates (BD Difco) and supplemented with 1% hemoglobin (Oxoid, Basingstoke, UK) and 1% IsoVitalex (BD BBL). MIC was determined in duplicates in conical 96-well plates (Sarstedt).

Colonies grown overnight on GC agar were suspended and diluted to a final optical density of 0.001, corresponding to 10^6^ CFU/mL in modified fastidious broth [39,40]; Difco Columbia Broth Base (BD) was supplemented with 15 mg/L hematin 0.05% Tween 80.6 mg/L and 15 mg/L of β-NAD (supplements from Sigma-Aldrich). Compounds were tested in 200 µL volume in two-fold dilutions with vehicle (2.5% DMSO) as the growth control. After 48 h incubation at 37 °C (5% CO_2_), 96-well plates were visually inspected for bacterial growth to determine MIC and 100 µL culture was plated on GC agar and incubated another 24 h to determine MBC. Selected compounds were also tested with 200 µM Fe (III) citrate in the broth.

#### 3.3.3. Susceptibility Testing for ESKAPE Pathogens

The MIC of the tested compounds was determined by the broth microdilution method in cation-adjusted Mueller Hinton broth (MHc, Becton Dickinson, Stockholm, Sweden). The susceptibilities of the six molecules, including ciprofloxacin (reference), was achieved according to Clinical and Laboratory Standards Institute 2020 2018 recommendations (CLSI. Methods for Dilution Antimicrobial Susceptibility Tests for Bacteria That Grow Aerobically. 11th ed. CLSI standard M07. Clinical and Laboratory Standards Institute; 2018). Briefly, from the subculture on non-selective agar medium of reference strains of *S. aureus* (ATCC^®^25923 and ATCC^®^700699), *E. faecalis* UCN41, *E. faecium* (BM4147, ATCC^®^19434T), *E. coli* (ATCC^®^25922), *K. pneumoniae* ATCC^®^700603, *P. aeruginosa* PAO1 and *A. baumannii* CIP7010, a bacterial suspension was elaborated to achieve a turbidity equivalent to the 0.5 McFarland turbidity standard. Then, the suspension was diluted 1:100 in MHc and 50 μL was used to inoculate the wells of a microplate to obtain 5.10^4^ CFU/wells (5.10^5^ CFU/mL). The compounds were directly diluted in MHc and transferred to the wells for a final volume of 100 μL. After 16 h incubation at 35 °C, the MIC was determined in comparison with the growth control wells. The measurements were performed in triplicate.

## 4. Conclusions

We reported here the synthesis and the antibacterial activity of new modified fluoroquinolone derivatives of ciprofloxacin. This new family of compounds presents the highest antibacterial activity against *C. trachomatis* with a potency on the nanomolar range for some derivatives. The activity against *N. gonorrhoeae* and ESKAPE pathogens is, however, lower than that of ciprofloxacin. The most potent compound against *C. trachomatis* has the greatest capacity to complex iron and its activity is markedly reduced in the presence of excess Fe (III). Iron chelation and the limitation of intracellular iron by this compound may, therefore, enhance its antichlamydial activity.

## Data Availability

Not applicable.

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
