# Peer review of "Modified Fluoroquinolones as Antimicrobial Compounds Targeting Chlamydia trachomatis"

_ijms, 2022, doi:10.3390/ijms23126741_

Round 1

Reviewer 1 Report

Dear authors, 

After reading your manuscript I have concluded the following:

1. the Introduction section is limited. Please include 1-2 more paragraphs providing information about the importance and novelty of your work.

2. In supplementary material please provide at least the 1HNMR spectrum as proof of the synthesis.

3. Proofreading is needed for minor spelling and grammar mistakes.

4. Can you please explain better the concept of structure modification? I mean the thought behind the structural modification of the compounds.

5. In general the manuscript is well written , tables and figures are in good quality and references are relevant to the subject. 

I do recommend publication with after minor revision, as the manuscript is relevant to the journal's scope. 

Best regards,

Author Response

Dear Colleague.

As you will read it below, we  addressed all your comments:

1. "the Introduction section is limited. Please include 1-2 more paragraphs providing information about the importance and novelty of your work": We included more information in the introduction to explain why we made the structural modifications we propose in this article, in particular, in relation to the iron requirements of the bacterium and the possibility to inhibit it.

2. "In supplementary material please provide at least the 1HNMR spectrum as proof of the synthesis": 1H and 13C NMR spectra were added at the end of the SI section.

3. "Proofreading is needed for minor spelling and grammar mistakes": we corrected all the mistakes with the help of a native English-speaking colleague.

4. "Can you please explain better the concept of structure modification? I mean the thought behind the structural modification of the compounds" : an explanation of the rational justifying the chemical modifications undertaken in this work has been added in the introduction part.

All changes made appear in red in the manuscript.

Best regards,

N. Serradji

Reviewer 2 Report

This manuscript describes the synthesis of new fluoroquinolones analogues and their antimicrobial targeting chlamydia trachomatis activity. The authors described their results properly. It seems this work is interesting and could be publishable in IJMS. However, to improve the quality of the manuscript authors, need to address the below comments.

1). Authors could change the scheme-1 title as a synthesis of compound 5a-d.

2). Authors need to refine Scheme-2. For example, compound numbers and ArCO2H could be included in the scheme. 

3). There are several typo errors in supporting information. For example, in 2b experimental procedure in Ethyl bromide E must be small later, and similar mistakes in 2c and 2d. In the experimental procedure, 2c IR is repeated. There are no compound numbers in many experimental procedures.   

4). Authors should provide C-F coupling constant for all compounds.

5). There are no NMRs in sporting information. Authors must provide 1H NMR and 13CNMRs for all compounds.

5). There are no melting points in the supporting information authors could provide complete chaptalization data for all new compounds.

Author Response

Dear colleague,

As you will read it below, we addressed all your comments:

1) "Authors could change the scheme-1 title as a synthesis of compound 5a-d" we modified the title into "Synthesis of compound 5a-d."

2) "Authors need to refine Scheme-2. For example, compound numbers and ArCO2H could be included in the scheme" : we corrected this scheme with the addition of "ArCOOH" and the compound numbers 7-12 (in the scheme and in its title). 

3)."There are several typo errors in supporting information. For example, in 2b experimental procedure in Ethyl bromide E must be small later, and similar mistakes in 2c and 2d. In the experimental procedure, 2c IR is repeated. There are no compound numbers in many experimental procedures" : we corrected many typo errors as well as the missing compound numbers in the SI section.

4) "Authors should provide C-F coupling constant for all compounds" 1JCF and 2JCF coupling constants were added when available. For greater convenience in understanding, zoomed windows have been added at the end of the SI section to illustrate these couplings.

5) "There are no NMRs in suporting information. Authors must provide 1H NMR and 13CNMRs for all compounds": 1H and 13C NMR spectra of all new compounds were added and the end of the SI section.

6) "There are no melting points in the supporting information authors could provide complete chaptalization data for all new compounds": melting points were measured and added for the new compounds.

All changes made appear in red in the attached manuscript, which also includes the SI section.

Best regards,

N. Serradji

Round 2

Reviewer 2 Report

In the revised manuscript authors addressed all the comments. This manuscript can be published in IJMS. However, while submitting the manuscript authors could address the below comments. 

1).  Authors must show the ArCO2H and need to provide the compound numbers Scheme 2.

2). Similarly, what is the compound number for ofloxacin in Figure 4. 

Author Response

Dear colleague,
I must admit that I do not understand the request of reviewer 2 because: - the numbers of the compounds are already indicated in Scheme 2, in bold, next to the structure where Ar is specified. However, to make it easier to read, the number of the compounds has been placed before the nature of Ar. - ofloxacin has no number because it was neither synthesized in this work nor evaluated for its biological activity, the data having been taken from the literature.
We included the structure of this compound in the text to illustrate the structural similarity with compound 16, tested in our study.
Best regards,
N. Serradji